# The Role of Interleukin-13 in Chronic Airway Diseases: A Cross-Sectional Study in COPD and Asthma–COPD Overlap

**DOI:** 10.3390/diseases13090287

**Published:** 2025-09-01

**Authors:** Marina Perković, Vesna Vukičević Lazarević, Pavo Perković, Tomislav Perković, Vanja Dolenec, Ana Hađak, Vesna Šupak Smolčić, Ljiljana Bulat Kardum

**Affiliations:** 1Special Hospital for Pulmonary Diseases, Rockefeller’s Street 3, 10000 Zagreb, Croatia; mpti2007@gmail.com (M.P.); vanjadolenec12@gmail.com (V.D.); hadjakana@gmail.com (A.H.); 2Clinic for Gynecological Diseases and Obstetrics, Clinic of Internal Medicine, University Clinic Merkur Zagreb, Zajec’s Street 19, 10000 Zagreb, Croatia; pavo.perkovic@zg.t-com.hr (P.P.); tomislavperkovic95@gmail.com (T.P.); 3Department of Laboratory Diagnostics, Clinical Hospital Center Rijeka, Kresimir’s Street 41, 51000 Rijeka, Croatia; vesnasupak@gmail.com (V.Š.S.); bulatkardum.ljiljana@gmail.com (L.B.K.)

**Keywords:** COPD, ACO, interleukin-13, endotype, phenotype, treatable traits

## Abstract

Background: Distinguishing chronic obstructive pulmonary disease (COPD) from asthma–COPD overlap (ACO) remains challenging due to shared clinical and inflammatory features. Interleukin-13 (IL-13) is implicated in airway inflammation and remodeling and may represent a potential treatable trait. This study aimed to evaluate whether serum IL-13 could differentiate between COPD and ACO or define ACO subtypes and to explore its relationship with clinical and phenotype parameters. Materials and Methods: We conducted a cross-sectional bicentric study in 215 COPD and ACO patients recruited from outpatient clinics. The study measured blood IL-13 levels in COPD vs. ACO patients, across five ACO subtypes, and evaluated IL-13’s ability to predict ACO. Additionally, correlations were explored among endotype (IL-13) and different phenotype traits (e.g., fractional exhaled nitric oxide (FeNO), sputum eosinophilia, serum total immunoglobulin E (tIgE) levels, blood eosinophilia, and neutrophilia) and clinical outcomes (annualized exacerbation rate, symptom scores, and pulmonary function parameters). Results: No significant differences in IL-13 levels were found between COPD and ACO patients or among ACO subtypes. IL-13 did not predict ACO occurrence. We observed a weak correlation between IL-13 and tIgE levels in the entire cohort. Additionally, there was a weak correlation between IL-13 and FeNO in patients with eosinophil counts exceeding 300 cells/μL, as well as between IL-13 and age in the COPD cohort. No correlation was found between IL-13 and other phenotypic features or clinical outcomes in the overall cohort, including within both COPD and ACO groups. Conclusions: IL-13 cannot differentiate between COPD and ACO or ACO’s subtypes.

## 1. Introduction

Chronic inflammatory airway diseases, such as asthma and chronic obstructive pulmonary disease (COPD), are driven by diverse underlying immunopathological mechanisms [1,2,3,4]. Asthma is predominantly associated with T helper 2 (Th2) lymphocyte activation and eosinophilic inflammation, whereas COPD typically involves Th1-mediated responses and neutrophilic inflammation [5]. However, this classical dichotomy does not capture the full spectrum of immune responses in COPD, as an estimated 10–40% of patients exhibit a mixed inflammatory profile in response to environmental and endogenous triggers [6,7]. This is often characterized by elevated eosinophil counts in induced sputum (>3%) or peripheral blood (>150 cells/μL) [8,9]. Such overlap complicates the clinical differentiation between asthma, COPD, and asthma–COPD overlap (ACO)—a heterogeneous condition encompassing several phenotypes likely driven by multiple inflammatory pathways [10,11]. Despite its clinical relevance, ACO lacks a universally accepted definition; current Global Initiative for Chronic Obstructive Lung Disease (GOLD) and Global Initiative for Asthma (GINA) guidelines offer only descriptive criteria without formal diagnostic consensus [12,13]. To advance the precision of diagnosis and treatment in chronic airway diseases, there is a critical need for reliable biomarkers and refined classification systems. Incorporating individual phenotypic and endotype traits and identifying treatable traits may enable more personalized and effective therapeutic strategies [14,15,16].

Phenotypic features, such as fractional exhaled nitric oxide (FeNO), serum total immunoglobulin E (tIgE) levels, and eosinophil counts in blood and sputum, are central to the concept of treatable traits and are increasingly used to guide diagnosis, therapeutic decisions, and disease monitoring in obstructive airway diseases [11,17,18].

FeNO is a robust indicator of T2 inflammation, primarily elevated in asthma and ACO, with lower levels in COPD [19]. FeNO levels are significantly (*p* < 0.001) lower in ACO compared to asthma (standardized mean difference (SMD) = −0.23) but significantly higher in ACO vs. COPD (SMD = 0.59). For asthma diagnosis sensitivity ranges approximately from 43% to 88% [20]. Specificity ranges from 60% to 92%, with specificity being higher than sensitivity in ruling in asthma or ACO [20]. FeNO > 31.5 ppb distinguished ACO from COPD (sensitivity 70%, specificity 90%) [19].

Both asthma and ACO may present with elevated serum IgE, generally higher in ACO than COPD (SMD = 0.42, *p* < 0.011) but hard to use for pure discrimination between ACO and asthma (SMD = 0.16, *p* < 0.096) [21].

Sputum eosinophils are significantly (*p* < 0.001) higher in asthma and ACO than in COPD (SMD = 0.62 for ACO vs. COPD), with no significant difference between ACO and asthma (SMD = −0.16, *p* = 0.23).

Blood eosinophils are elevated in both asthma and ACO and moderately elevated in some COPD phenotypes [9]. However, blood eosinophil counts are significantly higher in ACO compared to pure COPD, but not statistically different between ACO and asthma [21]. This limits their specificity for distinguishing ACO from asthma (SMD) = −0.08, *p* < 0.143), though they are useful (SMD = 0.44, *p* < 0.001) in distinguishing eosinophilic COPD or ACO from pure COPD [21]. In COPD, elevated blood eosinophil counts have been recognized as predictors of favorable response to inhaled corticosteroids (ICSs) and, more recently, to biologic therapies targeting interleukin-4 and interleukin-13 (IL-4/13) [22,23,24,25].

In parallel, endotype traits, defined by underlying cellular and molecular mechanisms, are gaining importance in the context of precision medicine [26,27]. Among these, alpha-1 antitrypsin (AAT) deficiency remains the most established endotype biomarker and treatable trait in COPD, with serum AAT levels serving to identify patients eligible for augmentation therapy [28].

Some emerging data suggest that high serum periostin and YKL-40 may be supportive for ACO, but these markers have not yet demonstrated strong routine clinical value due to inconsistent results and limited assay standardization [21,29].

Interleukin-13 (IL-13), along with IL-4, is a key mediator of Th2-driven airway inflammation [30]. It induces bronchial hyperactivity, promotes mucus production, IgE synthesis, and airway remodeling, and may also contribute to emphysema progression through eosinophil-mediated tissue damage [30,31]. Clinical evidence supports the efficacy of IL-4/IL-13 pathway inhibition in reducing exacerbations in both asthma and COPD [27,32]. In asthma, anti-IL-13 therapies have been associated with improved lung function, particularly in patients with peripheral eosinophil counts ≥ 300 cells/μL and elevated sputum IL-13 levels [33]. Similarly, in COPD patients with blood eosinophil counts exceeding 300 cells/μL, a threshold often used to define ACO, treatment with dupilumab, a monoclonal antibody targeting IL-4/IL-13 signaling, resulted in fewer exacerbations, better lung function, improved quality of life, and reduced symptom burden compared to placebo [27]. Given the central role of Th2 inflammation in chronic airway diseases, it is plausible that patients with elevated Th2 biomarkers, such as increased blood eosinophil counts, may also have higher serum IL-13 levels, potentially representing a treatable trait analogous to low AAT in COPD [34]. Taken together, these limitations highlight the need for additional biomarkers with improved specificity, reproducibility, and practicality in clinical practice. In this context, IL-13 warrants further investigation, particularly when considered alongside other T2 markers such as blood eosinophils and FeNO, to enhance the differentiation of ACO from COPD and to support a treatable-trait-based management approach.

This study investigated COPD and ACO by assessing a range of phenotypic and endotype traits alongside clinical outcomes. Building on prior research suggesting a role for IL-13 in differentiating asthma from COPD [35], the primary objective was to determine whether IL-13 levels could distinguish COPD from ACO and discriminate among ACO subtypes, thereby identifying patients who might benefit from IL-13-targeted therapy. Secondary aims included exploring correlations between IL-13 and other phenotypic markers, as well as clinical parameters such as annualized exacerbation rate (AER), symptom scores, and pulmonary function. Analyses were performed in the overall cohort, in patients with blood eosinophil counts > 300 cells/μL, and separately within the COPD and ACO groups. We hypothesized that IL-13 levels would be significantly higher in ACO patients—and in specific ACO subtypes—compared to those with COPD, and that IL-13 would correlate with clinical and phenotypic traits across all analytic groups.

## 2. Materials and Methods

### 2.1. Patient Enrollment and Eligibility Criteria

This bicentric non-interventional, observational, cross-sectional study was conducted at the Clinical Hospital Center Rijeka and the Special Hospital for Pulmonary Diseases in Zagreb, Croatia, following ethical approval from both institutional review boards (protocol numbers: 02-71/2021 and 2170-29-02/1-21-2). Sample size was determined through power analysis to ensure the study was adequately powered to detect differences in IL-13 levels and associations with clinical outcomes. Drawing on prior studies [35,36], a minimum of 200 participants was estimated to achieve a statistical power of 0.80 with a significance level of α = 0.05, allowing for subgroup analyses. Ultimately, data from 215 patients were analyzed, including 126 with COPD and 89 with ACO, ensuring sufficient statistical robustness.

### 2.2. Diagnostic Procedures

Phenotypic features assessed in this study included fractional exhaled nitric oxide (FeNO), peripheral blood eosinophil and neutrophil counts, sputum eosinophil presence, and total serum IgE levels. The primary endotypic marker examined was the concentration of IL-13 in peripheral blood. Table 1 provides a comprehensive overview of all phenotype and endotype traits measured, along with their respective methodologies and units. Each IL-13 sample was analyzed once due to budget limitations, as the study focused on comparing values between groups and assessing correlations with other parameters rather than determining absolute concentrations.

Clinical outcomes assessed in this study included the annualized exacerbation rate (AER), symptom burden evaluated using the COPD Assessment Test (CAT) questionnaire [37] and the modified Medical Research Council (mMRC) dyspnea scale [38], and disease severity classified by post-bronchodilator forced expiratory volume in one second (FEV_1_) percentage according to GOLD stages I–IV [12] Pulmonary function parameters were obtained using SensorMedics equipment (PFT System, VMAX229, CRCA 2000, Front, Italy) and included post-bronchodilator FEV_1_, forced vital capacity (FVC), FEV_1_/FVC ratio, diffusion capacity for carbon monoxide (DLCO), carbon monoxide transfer coefficient (KCO), and bronchodilator test (BDT), defined as an increase of ≥12% from baseline FEV_1_ after administration of four puffs of salbutamol.

Patients aged 40 years and older were recruited from outpatient pulmonary clinics at both institutions during routine clinical visits, provided they were in a stable phase of disease (no exacerbation in the preceding month). Data collection was conducted between February 2022 and March 2023. Pulmonologists obtained demographic data and medical histories, administered CAT and mMRC assessments, and performed anthropometric measurements (height and weight) to calculate body mass index (BMI) as weight (kg) divided by height squared (m^2^). Eligibility criteria were based on GOLD and GINA recommendations for COPD and ACO diagnoses [12,13]. Participants were randomly selected from scheduled appointments at the outpatient clinic for obstructive lung diseases. Inclusion and exclusion criteria are detailed in Table 2.

Patients were stratified according to GOLD stages, and subgroup analyses across GOLD I–IV classifications [12] were performed to assess whether IL-13 levels differed by the severity of airflow limitation in the overall cohort, as well as within the COPD and ACO groups. Additionally, patients with ACO were stratified into five distinct subgroups based on clinical, functional, and laboratory criteria: (1) ACO with a confirmed positive bronchodilator test; (2) ACO with a diagnosis of asthma before the age of 40; (3) ACO with peripheral blood eosinophil counts > 300 cells/μL; (4) ACO without identifiable phenotypic traits; and (5) ACO presenting with two or more of the aforementioned features.

### 2.3. Statistical Analysis

Data analysis was performed using SPSS version 25. The Kolmogorov–Smirnov test was used to assess the normality of numerical variables. Normally distributed data are presented as mean ± standard deviation, while non-normally distributed data are reported as median and range. Group comparisons were conducted using the chi-square test for categorical variables, analysis of variance (ANOVA) for normally distributed continuous variables, and the Kruskal–Wallis or Mann–Whitney U test for non-normally distributed variables. Correlation analyses were performed using Pearson’s or Spearman’s correlation coefficients, depending on data distribution. The predictive value of IL-13 for distinguishing ACO from COPD was evaluated using logistic regression analysis. A *p*-value of <0.05 was considered statistically significant.

## 3. Results

### 3.1. Study Population and Baseline Characteristics

Despite engaging 300 patients, 73 were excluded due to exclusion criteria, and 12 had incomplete data, as illustrated in the flow diagram depicted in Figure 1.

Data from 215 patients were analyzed, comprising 115 females (53.5%) and 100 males (46.5%). Of the participants, 126 patients (58.6%) were diagnosed with COPD, while 89 patients (41.4%) had ACO. There were no significant differences in gender (*p* = 0.50) or age (*p* = 0.45) distributions between the COPD and ACO groups. However, the COPD cohort exhibited significantly more years of smoking without differences in current smoking status (*p* = 0.012). On the other hand, the ACO cohort had a significantly higher BMI (*p* = 0.005), and 22.5% had been diagnosed with asthma before the age of 40, while there were no such patients in the COPD group. Demographic differences between the COPD and ACO groups are presented in Table 3.

### 3.2. Differences in Treatment Patterns

Furthermore, significant differences in treatment patterns were observed between ACO and COPD groups. ACO patients were more likely to use anti-inflammatory therapies, including ICS combinations (ICS/LABA/LAMA and ICS/LABA) and antileukotrienes. In contrast, COPD patients demonstrated greater reliance on oxygen therapy, LAMA monotherapy, and SABA. No significant differences were found in the use of LABA monotherapy, LABA/LAMA combinations, or SAMA. Further details are depicted in Figure 2.

### 3.3. Clinical Outcomes and Functional Assessments

Although there was no difference in CAT and mMRC scores between the two groups (*p* = 0.18 and *p* = 0.055), indicating similar levels of symptoms and dyspnea perception, the COPD group exhibited significantly worse pulmonary function test results (FEV1 *p* < 0.001, FEV1/FVC *p* <0.001, DLCO *p* < 0.001, and KCO *p* = 0.001) and AER (*p* = 0.044) as presented in Table 4.

### 3.4. Inflammatory Biomarkers and Phenotypic Traits

As expected, more patients in the ACO group had a positive bronchodilation test (*p* < 0.001) and higher levels of T2 inflammation markers (FeNO, *p* = 0.019; eosinophil blood cell count, *p* < 0.001; eosinophils in sputum, *p* < 0.001). However, there was no significant difference in total IgE (*p* = 0.278) and IL-13 (0.167) levels between the two groups. Disparities in outcomes and phenotype and endotype traits between COPD and ACO groups are presented in Table 5. Additionally, subgroup analyses across GOLD I–IV stages showed no statistically significant differences in IL-13 levels in the overall cohort (*p* = 0.7973). Consistently, no significant differences were detected when COPD (*p* = 0.4956) and ACO (*p* = 0.8237) groups were analyzed separately.

### 3.5. Correlation Analyses in the Total Cohort

By exploring correlations throughout the whole sample, we identified a noteworthy yet weak correlation (r = 0.171, *p* = 0.012) between IL-13 and IgE levels. However, our analysis revealed no significant correlations of IL-13 with other markers of T2 inflammation (e.g., eosinophil blood count and FeNO), symptoms (CAT, mMRC, AER), or lung function parameters (such as FEV1, FEV1/FVC ratio, FVC, BDT, DLCO, KCO) and disease severity, as depicted in Figure 3.

However, FeNO exhibited a weak negative correlation (r = −0.137, *p* = 0.044) with symptoms assessed by the CAT questionnaire and a weak positive correlation (r = 0.163, *p* = 0.002) with eosinophil blood count. Furthermore, FeNO demonstrated weak positive correlations with pulmonary function parameters, including DLCO and KCO, as illustrated in Figure 3.

### 3.6. Subgroup Analyses by Disease Type

While investigating the COPD and ACO groups separately, we did not identify a significant relationship between IL-13 and IgE levels. However, notable findings emerged; a slight positive correlation was observed between IL-13 levels and age (r = 0.289, *p* < 0.001) in the COPD group, and FeNO levels exhibited a weak negative correlation with smoking (r = −0.207, *p* = 0.020). In the ACO group, FeNO was weakly correlated with lung function (KCO) (r = 0.225, *p* = 0.034) and eosinophil blood count (r = 0.211, *p* = 0.047).

### 3.7. Findings in Patients with High Eosinophil Counts

Within the entire sample, 29 patients displayed eosinophil blood levels exceeding 300. Our study revealed a mild correlation (r = 0.369, *p* = 0.049) between IL-13 and FeNO levels and a mild to moderate correlation (r = 0.433, *p* = 0.019) between FeNO levels and lung function (DLCO).

### 3.8. Multivariate Analysis of IL-13 as Predictor of ACO

Multivariate logistic regression showed that IL-13 levels were not significantly associated with ACO. However, sex, DLCO (%), KCO (%), and a positive bronchodilator test emerged as statistically significant predictors of ACO, as shown in Table 6.

### 3.9. Analysis of ACO Subgroups

Furthermore, we stratified our ACO group into five distinct subgroups utilizing a range of clinical assessments, lab tests, and pulmonary function analyses: Group 1 comprised 20 patients with a positive bronchodilator test, Group 2 included 9 individuals diagnosed with asthma before the age of 40, Group 3 encompassed 17 patients with eosinophil counts above 300 cells per mcL, Group 4 consisted of 26 participants lacking specific phenotype characteristics, and Group 5 involved 17 subjects exhibiting more than one phenotype trait.

Upon comparing these five groups with the COPD group, we observed no significant differences in IL-13 levels (*p* = 0.317). However, a notable disparity was found for IgE levels (*p* = 0.034), with the highest levels observed in ACO Group 2. Additionally, higher levels of eosinophils in sputum (*p* < 0.001), particularly in ACO Groups 5 and 3, along with elevated eosinophil counts in peripheral blood (*p* < 0.001), underscored distinctive features within the ACO cohort.

## 4. Discussion

This study provides a comprehensive comparison of clinical, phenotype, and endotype characteristics between patients with COPD and those with ACO, contributing to a better understanding of their overlapping and distinct features. While several expected differences between the groups were confirmed, our primary hypotheses, particularly those related to IL-13, were not supported. Data from 215 patients were analyzed, with no significant differences observed in gender or age distribution between the COPD and ACO groups. However, variations in smoking duration, body mass index (BMI), and history of asthma diagnosis support the concept of ACO as a mixed phenotype [12,13]. This phenotype combines clinical features of both asthma and COPD, with obesity and early-onset asthma contributing to its distinct presentation [39,40,41].

Interestingly, patients with COPD showed significantly worse pulmonary function and higher annual exacerbation rates compared to those with ACO, suggesting more advanced airway obstruction and parenchymal damage [21]. This finding contrasts with previous studies, which generally report that ACO patients have more symptoms and poorer pulmonary function than those with asthma or COPD alone [39,40]. However, similar CAT and mMRC scores between the groups in our study suggest a comparable symptom burden and perception of dyspnea, highlighting that subjective symptoms do not always correlate strongly with objective measures such as lung function or exacerbation frequency [42]. As expected, ACO patients demonstrated a higher prevalence of positive bronchodilator responses, indicative of asthma-like reversible airflow limitation, and significantly elevated type 2 inflammatory markers, including blood and sputum eosinophils. They were also more likely to receive anti-inflammatory therapies.

However, no significant differences in IL-13 or total IgE levels were observed between the ACO and COPD groups or across GOLD I–IV stages. These findings are consistent with previous studies comparing asthma, ACO, and COPD, which also reported significant differences in certain phenotypic biomarkers between COPD and ACO, but not in IL-13 levels measured in peripheral blood [8] and sputum [36]. Interestingly, alternative omics-based approaches have shown greater discriminatory power between these diseases. A report demonstrated that NMR metabolic profiling of exhaled breath condensate (EBC) can differentiate asthma from COPD, even in smokers [43].

Moreover, in our study, even when the ACO group was stratified into five subgroups based on distinct phenotypic traits, IL-13 levels did not differ significantly among them, further underscoring the limited diagnostic utility of IL-13 in distinguishing between phenotypes within chronic airway diseases.

In the overall patient cohort, only a weak correlation was observed between IL-13 and total IgE levels, which is consistent with the known role of IL-13 in promoting IgE synthesis [30]. However, no associations were found between IL-13 and other T2 markers, symptoms, pulmonary function parameters, or disease severity. Furthermore, in the subgroup of patients characterized by eosinophil blood counts exceeding 300, a weak correlation was found between the endotype marker IL-13 and the phenotype biomarker FeNO. Weak correlations between IL-13 and other parameters, along with its lack of association with ACO in multivariate logistic regression, underscore its limited role as a standalone biomarker for characterizing ACO.

However, the principal component and network analysis in a previous study revealed a mixed inflammatory pattern in ACO, positioned between asthma and COPD [8]. Notably, IL-13 emerged as the most interconnected node in the network, with varying weights across the three conditions. The conclusion of the research was that asthma and COPD represent distinct inflammatory conditions that may intersect in certain patients, manifesting as a mixed inflammatory pattern, with IL-13 potentially playing a central role in regulating inflammation in these contexts [8]. Its utility may be greater in a subgroup of patients with elevated eosinophil blood counts, where we found correlation between IL-13 and FeNO. This subgroup of COPD and ACO patients suggests that IL-13 could serve as a treatable trait rather than a distinguishing factor between ACO and COPD. This is supported by studies involving anti-IL-4/13 antibody, which have shown the best outcomes in patients with high blood eosinophil counts [24,25].

Despite animal models demonstrating that eosinophilic IL-13, by inducing MMP-12, contributes to emphysema development [31], we did not find a correlation between IL-13 levels and low DLCO and KCO, indicative of emphysema [44].

The multivariate logistic regression analysis showed that a higher DLCO (%) was associated with an increased likelihood of ACO, whereas KCO (%) demonstrated a negative association. Higher DLCO in ACO patients likely reflects preserved alveolar–capillary membrane integrity and a lower degree of emphysematous destruction compared with COPD alone, consistent with the asthmatic component of ACO, where alveolar architecture is generally maintained [45,46]. In contrast, the reduced KCO, which represents DLCO normalized for alveolar volume (VA), suggests increased VA due to hyperinflation and heterogeneous ventilation—features more characteristic of asthma than pure COPD [47]. This interpretation aligns with previous studies showing that low KCO is associated with an increased risk of exacerbations in ACO [48]. Furthermore, in the multivariate logistic regression model, a positive bronchodilator test was a strong predictor of ACO, which is expected given that it is one of the diagnostic criteria used to distinguish ACO from COPD [12,13].

In our study, male sex was negatively associated with ACO, suggesting that female sex is more frequently linked to this phenotype. This finding is consistent with previous reports indicating a higher prevalence of ACO among women [49], likely reflecting the greater occurrence of adult-onset asthma in females, hormonal influences on airway inflammation, and increased airway hyperresponsiveness compared with males [50,51].

Furthermore, we were surprised to find a slight positive correlation between age and IL-13 levels in a group of COPD patients. We did not find similar observations in the literature regarding serum IL-13. However, higher IL-13 levels have been identified in the aqueous humor of patients with age-related macular degeneration [52]. This may be explained by the fact that inflammatory markers, including interleukins like IL-13, tend to increase with age. This phenomenon, known as “inflammaging”, refers to the chronic, low-grade inflammation commonly observed in older adults [35].

On the other hand, as expected, we also found a correlation between Th2 markers, FeNO, and blood eosinophil count. Furthermore, higher FeNO levels were associated with fewer symptoms (using a symptoms score like CAT) and better pulmonary function, aligning with results from similar studies in COPD patients [44]. However, this finding is confusing, as a higher blood eosinophil count is a criterion for defining ACO [12,13] and, as mentioned earlier, ACO patients compared to patients with COPD alone experience more symptoms and have poorer pulmonary function [39,40]. Furthermore, higher FeNO is associated with accelerated FEV1 decline in individuals with chronic airway disease in the general population [53]. However, there is a study [54] that supports our findings and like our study challenges the current understanding of the role of eosinophilic inflammation in chronic obstructive disease.

This study has several limitations that warrant consideration. First, the relatively small sample size (215 patients) and the recruitment from a limited number of centers may reduce statistical power, introduce selection bias, and limit the generalizability of the findings to broader populations. Second, the cross-sectional design precludes causal inference between IL-13 levels and the presence of ACO, as all measurements were obtained at a single time point. Third, due to resource constraints, IL-13 measurements were performed in single runs rather than triplicates, potentially increasing measurement variability. Standardizing biomarker assays, performing triplicate analyses, and applying uniform phenotype definitions in future studies could improve the reliability of IL-13 as a potential differentiating or endotype biomarker.

Future research should include larger, multicenter, longitudinal cohorts to confirm these associations, explore temporal changes in IL-13, and evaluate their predictive value for clinical outcomes such as exacerbations, disease progression, and response to personalized treatment strategies. While our data suggest that IL-13 alone may have limited diagnostic value due to its weak correlations with other Th2 markers and clinical parameters, its role in specific subgroups—such as patients with marked eosinophilia—or in combination with other biomarkers merits further investigation. Finally, the absence of mechanistic analyses limits our ability to interpret biological pathways underlying our findings, including the observed correlation between IL-13 and age in COPD patients. Integrating molecular approaches such as transcriptomics or proteomics in future studies could help elucidate links between IL-13, aging, and inflammation, particularly in the context of “inflammaging” [52].

## 5. Conclusions

This study contributes valuable insights into the heterogeneity of chronic obstructive diseases and the limitations of IL-13 as a biomarker. However further research is essential to address these limitations and refine our understanding of the complex interplay between endotype and phenotype traits and clinical outcomes in chronic airway diseases.

## Figures and Tables

**Figure 1 diseases-13-00287-f001:**
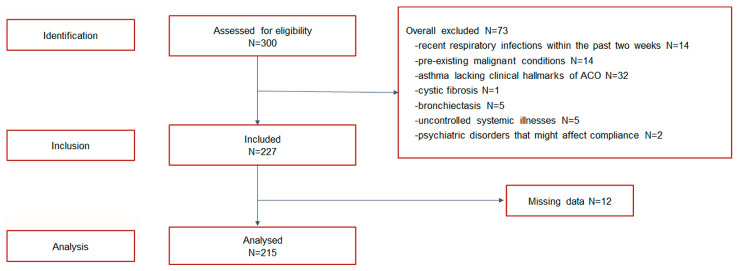
Flow diagram illustrating patient selection and data completeness in the study.

**Figure 2 diseases-13-00287-f002:**
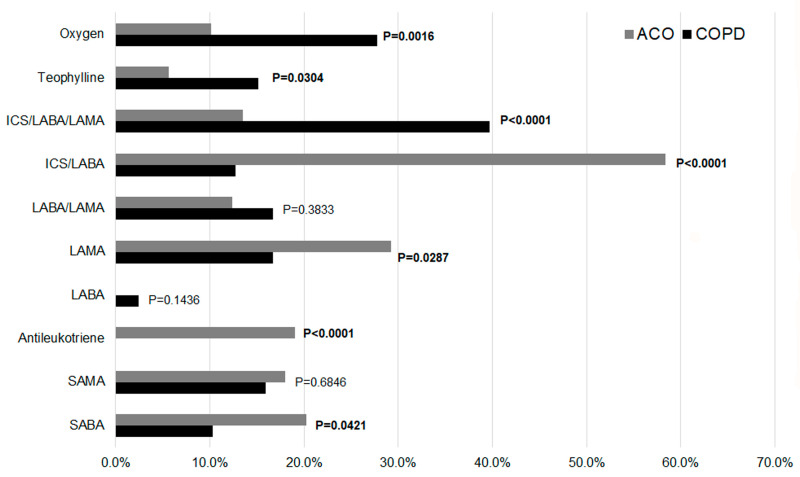
Differences in therapy between COPD and ACO groups.

**Figure 3 diseases-13-00287-f003:**
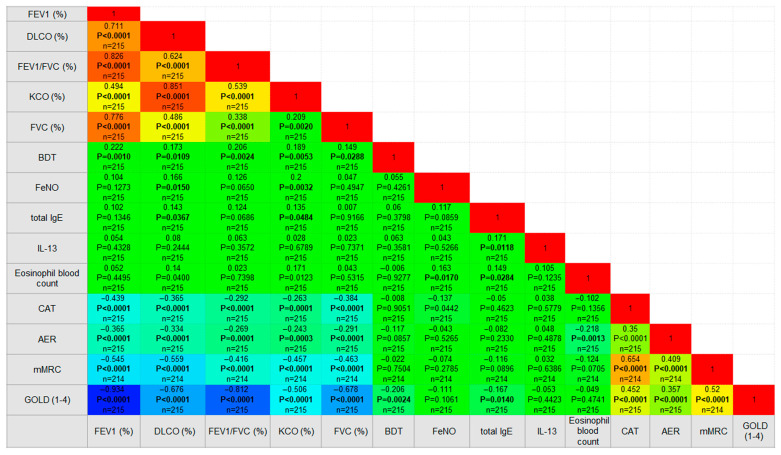
Correlelogram of measured variables (Spearman’s correlation). Colors indicate the strength and direction of correlation coefficients (*r*), ranging from strong positive (red) to no correlation (green) and strong negative (blue). Numbers in the cells represent correlation coefficients (*r*) with corresponding *p*-values and sample size (*n*).

**Table 1 diseases-13-00287-t001:** Detailed overview of phenotype and endotype features, including measurement methods and units.

Measurement	Equipment/Method	Measurement Units
FeNO	Medisoft Equipment (Sorrines, Belgium)	Ppb
Blood Eosinophils	Laser light scatter and myeloperoxidase cytochemical staining for leukocyte differential (ADVIA 2120, Siemens Healthineers, Erlangen, Germany))	Cells/mcl
Blood Neutrophils	Laser light scatter and myeloperoxidase cytochemical staining for leukocyte differential (ADVIA 2120, Siemens Healthineers, Erlangen, Germany))	Cells/mcl
Total IgE	UniCAP fluoroimmunoassay (ThermoFisher, Uppsala, Sweden)	IU/mL
Sputum Eosinophils	Cytology analysis by an experienced cytology technician	Presence/Absence
Interleukin-13	ELISA (BioVendor, Laboratory Medicine, Brno, Czech Republic). The method involved the use of microtiter plates, monoclonal antibodies targeting IL-13, biotin-labeled secondary antibodies, and spectrophotometric measurement of IL-13 concentration at a wavelength of 450 nm. The detection range was 1.6–100 pg/mL with a limit of detection of 0.7 pg/mL and an analytical imprecision of 6.0%	pg/mL

FeNO = Fractional Exhaled Nitric Oxide, IgE = immunoglobulin E.

**Table 2 diseases-13-00287-t002:** Inclusion and exclusion criteria of the study.

Criteria Type	Group	Criteria Description
Inclusion	COPD	Adults with a history of smoking and/or toxic exposure, low birth weight, or respiratory illness (e.g., tuberculosis) No past or current diagnosis of asthma Exhibiting persistent respiratory symptoms such as dyspnea, cough, or increased sputum production Spirometry showing post-bronchodilator FEV1/FVC ratio < 70%
	ACO	Diagnosed with COPD plus: Post-bronchodilator FEV1 increase > 12% or > 200 mL Asthma diagnosis before age 40 Blood eosinophil count > 300 cells/μL
Exclusion	Both COPD and ACO	Recent respiratory infections within the past two weeks Pre-existing malignant conditions Asthma lacking clinical hallmarks of ACO Diagnosed with cystic fibrosis or bronchiectasis Uncontrolled systemic illnesses Psychiatric disorders affecting compliance

COPD—Chronic obstructive pulmonary disease, ACO—asthma–COPD overlap, FEV1—forced expiratory volume in 1 s, FVC—forced vital capacity.

**Table 3 diseases-13-00287-t003:** Demographic comparison between COPD and ACO groups.

Variable	COPD N = 126	ACO N = 89	*p*-Value
Age mean (median, range)	70.5 (69.5. 48–88)	69.3 (70. 42–92)	0.460
Gender n (%) Females (F) Males (M)	65 (51.6%) 61 (48.4%)	50 (56.2%) 39 (43.8%)	0.507
BMI mean (median, range)	26.34 (25.94. 14.3–47.05)	28.25 (27.6. 18.22–45.7)	0.005
Smoking history Pack/years Mean ± SD	44.6 ± 22.6	31.0 ± 22.0	<0.001
Smoking history n (%) Never smoker Ex-smoker Active smoker	9 (7.1%) 66 (52.4%) 51 (40.5%)	14 (15.7%) 40 (44.9%) 35 (39.3%)	0.123
Asthma diagnosed before the age of 40 n (%)	0 (0%)	20 (22.5%)	<0.001

Parametric tests utilized for comparing values included one-way analysis of variance (ANOVA), while non-parametric tests involved Mann–Whitney tests. A *p*-value less than 0.05 is regarded as statistically significant. COPD—chronic obstructive pulmonary disease. ACO—asthma–COPD overlap.

**Table 4 diseases-13-00287-t004:** Disparities in Clinical and Functional Outcome between COPD and ACO group.

Variable	COPD N = 126	ACO N = 89	*p*-Value
Bronchodilator tests Positive n (%) Negative n (%)	10 (7.9%) 116 (92.1%)	34 (38.2%) 55 (61.8%)	**<0.001**
FEV1/FVC (%) Mean (median, range)	43.87 (43.6. 21–77%)	52.315 (54. 23–72)	**<0.001**
FEV1 (%) Median (range)	46 (17–114)	71 (18–158)	**<0.001**
GOLD grade of severity of postbronchodilatory airflow obstruction n (%) of patients I II III IV	21 (16.7) 28 (22.2) 50 (39.7) 27 (21.4)	32 (35.8) 23 (25.8) 29 (35.5) 5 (5.6)	**0.001**
DLCO (%) Mean (median, range)	55.0 (49.5. 12–114)	72.2 (71, 20–139)	**<0.001**
KCO (%) Mean ± SD	63.7 ± 23.9	75.5 ± 24.8	**0.001**
AER Mean (median, range)	0.70 (0, 0–8)	0.47 (0, 0–4)	**0.044**
CAT Mean (median, range)	18.2 (19, 1–34)	16.7 (18, 1–32)	0.181
mMRC (median, range)	2 (0–4)	2 (0–4)	0.055

Parametric tests utilized for comparing values included one-way analysis of variance (ANOVA), while non-parametric tests involved Mann–Whitney tests. A *p*-value less than 0.05 is regarded as statistically significant. COPD—chronic obstructive pulmonary disease, ACO—asthma–COPD overlap, FEV1—Forced expiratory volume in 1 s, FVC—Forced vital capacity, GOLD—Global Initiative for chronic obstructive lung disease, DLCO—diffusing capacity of the lungs for carbon monoxide, KCO—Carbon monoxide transfer coefficient, AER—annualized exacerbation rate, CAT—COPD Assessment Test, mMRC—Modified Medical Research Council Dyspnea Scale.

**Table 5 diseases-13-00287-t005:** Contrasts in Phenotype and Endotype traits between COPD and ACO groups.

Variable	COPD N = 126	ACO N = 89	*p*-Value
FeNO (ppb) Median (range)	18 (4–86)	21 (3–363)	**0.019**
Sputum eosinophils Positive n (%) Negative n (%)	0 (0) 126 (100)	37 (41.6) 52 (58.4)	**<0.001**
Eosinophil Blood Cell Count (cells/µL) Median (range)	135 (0–260)	180 (0–1170)	**<0.001**
Neutrophil Blood Cell Count (cells/µL) Median (range)	5265 (1180–16,370)	4760 (1700–13,280)	0.058
Total Immunoglobulin E (kIU/L) Median (range)	52.5 (2–3450)	63 (2–1607)	0.278
Interleukin-13 (pg/mL) Median (range)	0.1 (0–863)	0.5 (0–12,189)	0.167

Parametric tests utilized for comparing values included one-way analysis of variance (ANOVA), while non-parametric tests involved Mann–Whitney tests. A *p*-value less than 0.05 is regarded as statistically significant. FeNO—Fractional exhaled Nitric Oxide.

**Table 6 diseases-13-00287-t006:** A summary table of the multivariate logistic regression.

Variable	Coefficient	SE	Wald	*p*-Value	OR	95% CI for OR
Age (years)	0.0073835	0.0238782	0.09639	0.756	1.0074	0.9615–1.0555
Sex (male)	−0.97961	0.46385	4.4602	**0.035**	0.3755	0.1513–0.9320
BMI (kg/m^2^)	0.016913	0.043039	0.1544	0.694	1.0171	0.9348–1.1066
CRP (mg/L)	−0.014537	0.025343	0.329	0.566	0.9856	0.9378–1.0358
Blood neutrophils (cells/µL)	0.000033876	0.0001007	0.1132	0.736	1.00	0.9998–1.0002
Blood eosinophils (cells/µL)	0.0042856	0.002332	3.3772	0.066	1.0043	0.9997–1.0089
Sputum eosinophils (present)	23.181118	6322.15461	0.000013	0.997	1.17 × 10^10^	–
Total IgE (kIU/L)	0.00033132	0.00043649	0.5761	0.448	1.0003	0.9995–1.0012
Interleukin-13 (pg/mL)	0.001276	0.0012637	1.0195	0.313	1.0013	0.9988–1.0038
FeNO (ppb)	0.016517	0.010378	2.5331	0.1115	1.0167	0.9962–1.0375
DLCO (%)	0.061369	0.018344	11.1928	**0.001**	1.0633	1.0257–1.1022
KCO (%)	−0.041758	0.019225	4.7181	**0.030**	0.9591	0.9236–0.9959
Bronchodilator test (+)	2.6837	0.53691	24.9845	**<0.001**	14.6391	5.1108–41.9314
mMRC score	0.083014	0.22258	0.1391	0.709	1.086	0.7024–1.6808

Bold indicates statistical significance at *p* < 0.05. OR = Odds ratio; CI = Confidence interval; SE = Standard error; BMI = Body mass index; CRP = C-reactive protein; IgE = Immunoglobulin E; FeNO = Fractional exhaled nitric oxide; DLCO = Diffusing capacity for carbon monoxide; KCO = Carbon monoxide transfer coefficient; mMRC = Modified Medical Research Council Dyspnea Scale.

## Data Availability

The raw data supporting the conclusions of this article will be made available by the authors on request.

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
