# Peer review of "The Role of Interleukin-13 in Chronic Airway Diseases: A Cross-Sectional Study in COPD and Asthma–COPD Overlap"

_diseases, 2025, doi:10.3390/diseases13090287_

Round 1

Reviewer 1 Report

Comments and Suggestions for Authors

In this manuscript, Perković and colleagues investigate whether serum IL-13 levels can distinguish between patients with COPD and those with asthma–COPD overlap (ACO). The study is well-structured and clearly presented. However, several aspects could be improved to enhance the clarity and scientific rigor of the work.

1. One limitation of the current analysis is the lack of stratification or adjustment based on GOLD stages. The authors should consider performing subgroup analyses across GOLD I–IV classifications to investigate whether IL-13 levels differ based on airflow limitation severity. Such an approach could not only uncover potential associations between IL-13 and disease stage but also help explain the considerable variability in IL-13 levels observed within both the COPD and ACO groups.

2. The Methods section would benefit from clearer structure. The authors are encouraged to divide it into subheadings, such as: Patient enrollment and eligibility criteria; Diagnostic procedures (including the flow cytometry approach); Statistical analysis.

3. It is not clear what specific data were obtained via flow cytometry. No information is provided regarding the antibodies used, gating strategies, or whether immunophenotyping was performed. If flow cytometry was not antibody-based, this term should be revised.

4. The authors should clarify whether technical replicates (e.g., triplicates) were performed for IL-13 ELISA measurements to ensure reliability of the results.

5. The quality of the figures is suboptimal. The text within the figures is difficult to read, which limits interpretability. The authors should consider providing high-resolution versions or redesigning figures for clarity.

6. A summary table of the multivariate logistic regression analysis is missing. Including this would greatly improve the transparency and accessibility of the statistical results.

Author Response

Comments 1: One limitation of the current analysis is the lack of stratification or adjustment based on GOLD stages. The authors should consider performing subgroup analyses across GOLD I–IV classifications to investigate whether IL-13 levels differ based on airflow limitation severity. Such an approach could not only uncover potential associations between IL-13 and disease stage but also help explain the considerable variability in IL-13 levels observed within both the COPD and ACO groups.

Response 1: We thank the reviewer for this valuable suggestion. As recommended, we performed subgroup analyses across GOLD I–IV classifications to assess whether IL-13 levels differed according to airflow limitation severity. The methodology section has been updated accordingly (Methods, paragraph 2.2, lines 166–169), along with the Results (paragraph 3.4, lines 239–242) and Discussion (line 325). The subgroup analysis revealed no statistically significant differences in IL-13 levels across GOLD stages in the overall cohort (p = 0.7973). Likewise, when COPD (p = 0.4956) and ACO (p = 0.8237) groups were analyzed separately, no significant differences were found. 

Comments 2: The Methods section would benefit from clearer structure. The authors are encouraged to divide it into subheadings, such as: Patient enrollment and eligibility criteria; Diagnostic procedures (including the flow cytometry approach); Statistical analysis.

Response 2: We appreciate the reviewer’s suggestion to improve the clarity of the Methods section. As recommended, we have restructured this section into clear subheadings: Patient Enrollment and Eligibility Criteria (line 119), Diagnostic Procedures (including the flow cytometry approach, line 132), and Statistical Analysis (line 174). This change improves readability and ensures that each methodological component is clearly delineated.

Comments 3: It is not clear what specific data were obtained via flow cytometry. No information is provided regarding the antibodies used, gating strategies, or whether immunophenotyping was performed. If flow cytometry was not antibody-based, this term should be revised

Response 3: The flow cytometry was not antibody based it was done with ADVIA 2120 hematology analyzer (Siemens Healthcare Diagnostics); hydrodynamically focused laser light scatter and myeloperoxidase cytochemical staining for leukocyte differential. This was changed in Table 1.

Comments 4: The authors should clarify whether technical replicates (e.g., triplicates) were performed for IL-13 ELISA measurements to ensure reliability of the results.

Response 4: We appreciate the reviewer’s comment. Technical replicates were not performed for IL-13 ELISA measurements. This decision was made due to financial constraints and because our primary objective was not to determine the absolute IL-13 concentrations, but to compare values between experimental groups and to evaluate correlations with other measured parameters. Therefore, each sample was measured once, and statistical analyses focused on relative differences and associations rather than absolute quantification. We have clarified this point in the Methods section, paragraph 2.2 (Diagnostic Procedures, lines 137–139), and addressed it as a study limitation in the Discussion (lines 401–402).

Comments 5: The quality of the figures is suboptimal. The text within the figures is difficult to read, which limits interpretability. The authors should consider providing high-resolution versions or redesigning figures for clarity.

Response 5: The resolution of all three figures has been improved.

Comments 6: A summary table of the multivariate logistic regression analysis is missing. Including this would greatly improve the transparency and accessibility of the statistical results.

 Response 6: I have added Summary table- Table 6 and, in accordance with the results, revised the paragraph Multivariate Analysis of IL-13 as a Predictor of ACO section in the Results (lines 278–280) and the Discussion (lines 360–377) to highlight the statistically significant predictors.

Reviewer 2 Report

Comments and Suggestions for Authors

The paper is quite well written. The topic is current and interesting. I have some suggestions:

1) The study's sample size of 215 patients and its cross-sectional nature may limit the generalizability of the findings and preclude establishing causal relationships. Future research with larger, multicenter cohorts and longitudinal designs could provide more definitive insights into the role of IL-13 in differentiating COPD and ACO. Please, add some comments.

2) Variations in serum IL-13 measurement techniques and the criteria used to classify ACO subtypes may affect the accuracy of the results. Standardizing biomarker assays and phenotype definitions in future studies could enhance the reliability of IL-13 as a potential differentiating or phenotyping biomarker.

3) While the introduction highlights IL-13 and eosinophil counts as key biomarkers, it could benefit from a more comprehensive discussion of how these markers compare to other established or emerging biomarkers in differentiating COPD, ACO, and asthma. Clarifying the current gaps in biomarker specificity, sensitivity, or practicality would strengthen the rationale for focusing on IL-13 and justify its potential as a distinctive treatable trait.

4) 1. Introduction L33-35. Chronic inflammatory airway diseases, such as asthma and chronic obstructive pulmonary disease (COPD), are driven by diverse underlying immunopathological mechanisms [1,2].  Authors are kindly requested to emphasize the current concepts about these issues in the context of recent knowledge and the available literature. These articles should be quoted in the References list. References 1. Definition and Nomenclature of Chronic Obstructive Pulmonary Disease: Time for Its Revision. Am J Respir Crit Care Med. 2022 Dec 1;206(11):1317-1325. doi: 10.1164/rccm.202204-0671PP;    2. Chronic Obstructive Pulmonary Disease Definition: Is It Time to Incorporate the Concept of Failure of Lung Regeneration? Am J Respir Crit Care Med. 2023 Feb 1;207(3):366-367. doi: 10.1164/rccm.202208-1508LE.

5) Although the introduction outlines the study’s aims to evaluate IL-13 in distinguishing clinical phenotypes, it could more explicitly articulate how this research advances existing knowledge or addresses specific limitations of previous studies. Additionally, emphasizing the potential impact of identifying IL-13 as a biomarker on clinical decision-making, personalized therapy, or patient outcomes would enhance the significance and relevance of the study context.

6) Methods/results. While the study included 215 patients from two centers in Croatia, the relatively modest sample size and geographic concentration may limit the applicability of the findings to broader, more diverse populations. Future research should involve larger, multicenter cohorts with varied demographic and ethnic backgrounds to enhance the external validity of the results. Please, discuss these limitations.

7) Methods/results.The cross-sectional nature of this study precludes assessment of causal relationships and temporal changes in biomarkers such as IL-13 and disease progression. Longitudinal studies are warranted to evaluate how inflammatory markers evolve over time and their potential role in predicting disease exacerbations, treatment responses, or progression from COPD to ACO. Please, discuss these observations.

8) Discussion. The relatively modest sample size of 215 patients may limit the statistical power to detect subtle differences or associations, particularly within stratified subgroups such as those based on phenotypic traits or eosinophil counts. Future studies with larger, multicenter cohorts are necessary to validate these findings and ensure their generalizability across diverse populations. Please, ameliorate the discussion.

9) Discussion. The cross-sectional design of this study precludes evaluation of temporal changes in biomarkers like IL-13, FENO, and eosinophil levels. Longitudinal studies are required to determine the stability of these biomarkers over time, their predictive value for clinical outcomes such as exacerbations or disease progression, and their potential role in guiding personalized treatment strategies. Please, discuss this point in the paragraph.

Author Response

Comments 1: The study's sample size of 215 patients and its cross-sectional nature may limit the generalizability of the findings and preclude establishing causal relationships. Future research with larger, multicenter cohorts and longitudinal designs could provide more definitive insights into the role of IL-13 in differentiating COPD and ACO. Please, add some comments.

Response 1: Thank you for pointing this out. We agree with this comment. Therefore, in accordance with your guidance, we have revised the Discussion, specifically the paragraphs addressing the study’s limitations and the need for future research (lines 396–401 and 406-407). 

Comments 2: Variations in serum IL-13 measurement techniques and the criteria used to classify ACO subtypes may affect the accuracy of the results. Standardizing biomarker assays and phenotype definitions in future studies could enhance the reliability of IL-13 as a potential differentiating or phenotyping biomarker.

Response 2: Agree. We have, accordingly, we have modified the Discussion, specifically the paragraphs addressing the study’s limitations and the need for future research (lines 403–405) to emphasize this point.

Comments 3: While the introduction highlights IL-13 and eosinophil counts as key biomarkers, it could benefit from a more comprehensive discussion of how these markers compare to other established or emerging biomarkers in differentiating COPD, ACO, and asthma. Clarifying the current gaps in biomarker specificity, sensitivity, or practicality would strengthen the rationale for focusing on IL-13 and justify its potential as a distinctive treatable trait. 

Response 3: In response to the reviewer’s comment, we have expanded the Introduction (lines 56–74, 83–85, and 100–105) to provide a more comprehensive comparison of IL-13 and eosinophil counts with other established and emerging biomarkers for differentiating COPD, ACO, and asthma. We have also clarified the current gaps in specificity, sensitivity, and clinical practicality to strengthen the rationale for focusing on IL-13 as a potential distinctive treatable trait. 

Comments 4; 1. Introduction L33-35. Chronic inflammatory airway diseases, such as asthma and chronic obstructive pulmonary disease (COPD), are driven by diverse underlying immunopathological mechanisms [1,2].  Authors are kindly requested to emphasize the current concepts about these issues in the context of recent knowledge and the available literature. These articles should be quoted in the References list. References 1. Definition and Nomenclature of Chronic Obstructive Pulmonary Disease: Time for Its Revision. Am J Respir Crit Care Med. 2022 Dec 1;206(11):1317-1325. doi: 10.1164/rccm.202204-0671PP;    2. Chronic Obstructive Pulmonary Disease Definition: Is It Time to Incorporate the Concept of Failure of Lung Regeneration? Am J Respir Crit Care Med. 2023 Feb 1;207(3):366-367. doi: 10.1164/rccm.202208-1508LE. 

Response 4; In accordance with the reviewer’s suggestion, we have incorporated the recommended references into the Introduction (lines 34-36) to emphasize current concepts on chronic inflammatory airway diseases in the context of recent knowledge and available literature. The cited articles have been added to the References list as references [3] and [4]. 

Comments 5; Although the introduction outlines the study’s aims to evaluate IL-13 in distinguishing clinical phenotypes, it could more explicitly articulate how this research advances existing knowledge or addresses specific limitations of previous studies. Additionally, emphasizing the potential impact of identifying IL-13 as a biomarker on clinical decision-making, personalized therapy, or patient outcomes would enhance the significance and relevance of the study context.

Response 5; In response to the reviewer’s comment, we have revised the Introduction (lines 100–105) to more explicitly articulate how this study advances existing knowledge and addresses specific limitations of previous research. We have also emphasized the potential impact of identifying IL-13 as a biomarker on clinical decision-making, personalized therapy, and patient outcomes.

Comments 6: Methods/results. While the study included 215 patients from two centers in Croatia, the relatively modest sample size and geographic concentration may limit the applicability of the findings to broader, more diverse populations. Future research should involve larger, multicenter cohorts with varied demographic and ethnic backgrounds to enhance the external validity of the results. Please, discuss these limitations.

Response 6; In accordance with your guidance, we have revised the Discussion, specifically the paragraphs addressing the study’s limitations and the need for future research (lines 396–401 and 406-407).

Comments 7: Methods/results.The cross-sectional nature of this study precludes assessment of causal relationships and temporal changes in biomarkers such as IL-13 and disease progression. Longitudinal studies are warranted to evaluate how inflammatory markers evolve over time and their potential role in predicting disease exacerbations, treatment responses, or progression from COPD to ACO. Please, discuss these observations.

Response 7; In accordance with your guidance, we have revised the Discussion, specifically the paragraphs addressing the study’s limitations and the need for future research (lines 396–401 and 406-407).

Comments 8; Discussion. The relatively modest sample size of 215 patients may limit the statistical power to detect subtle differences or associations, particularly within stratified subgroups such as those based on phenotypic traits or eosinophil counts. Future studies with larger, multicenter cohorts are necessary to validate these findings and ensure their generalizability across diverse populations. Please, ameliorate the discussion.

Response 8; in accordance with your guidance, we have revised the Discussion, specifically the paragraphs addressing the study’s limitations and the need for future research (lines 396–401 and 406-407).

Comments 9; Discussion. The cross-sectional design of this study precludes evaluation of temporal changes in biomarkers like IL-13, FENO, and eosinophil levels. Longitudinal studies are required to determine the stability of these biomarkers over time, their predictive value for clinical outcomes such as exacerbations or disease progression, and their potential role in guiding personalized treatment strategies. Please, discuss this point in the paragraph.

Response 9; In accordance with your guidance, we have revised the Discussion, specifically the paragraphs addressing the study’s limitations and the need for future research (lines 399–401 and 406-407). 

Round 2

Reviewer 1 Report

Comments and Suggestions for Authors

The authors have addressed all the questions and improved the manuscript. I believe the manuscript can be accepted in its current form.

Reviewer 2 Report

Comments and Suggestions for Authors

The paper has been improved, as requested. No further comments.